# Prevalence of Selected Single-Nucleotide Variants in Patients with Neuroendocrine Tumors—Potential Clinical Relevance

**DOI:** 10.3390/jcm11195536

**Published:** 2022-09-21

**Authors:** Anna Kurzyńska, Dorota Pach, Anna Elżbieta Skalniak, Agnieszka Stefańska, Marta Opalińska, Elwira Przybylik-Mazurek, Alicja Hubalewska-Dydejczyk

**Affiliations:** 1Chair and Department of Endocrinology, Jagiellonian University Medical College, 30-688 Kraków, Poland; 2Department of Endocrinology, Oncological Endocrinology and Nuclear Medicine, University Hospital, 30-688 Kraków, Poland

**Keywords:** neuroendocrine neoplasms, single-nucleotide polymorphisms, molecular markers

## Abstract

Introduction: The genetic basis of neuroendocrine tumors (NETs), whose incidence is continuously increasing, is still not fully defined. The majority of NETs are sporadic, and only a small percentage occur as part of hereditary genetic syndromes. However, the associations of multiple genetic variants have been found as clinically relevant in several neoplasms. The aim of this study was to evaluate whether selected, literature-based genetic variants may have a potential role in NET susceptibility and clinical outcome in Polish patients. Materials/methods: A total of 185 patients recruited from one clinical center were enrolled. In the first part of the study, the molecular analysis including four single-nucleotide variants (rs8005354 (*DAD1*, NM_001344 intronic T/C substitution), rs2069762 (T/G substitution in the promoter region of the *IL2* NM_000586), rs3731198 (*CDKN2A*, NM_000077 intronic A/G substitution), and rs1800872 (C/A substitution in the promoter region of the *IL10* NM_000572)) was performed in 107 participants (49 patients with NETs with different primary site NETs and a control group of 58 healthy adult volunteers). In the second stage, the same single-nucleotide polymorphisms (SNPs) were assessed in 127 patients with NET and analyzed in terms of clinical data (primary site, serum CgA concentration, and metastatic disease). Results: The analysis of homozygotes revealed a statistically significant higher prevalence of TT homozygotes of variant rs3731198 in the control group (*p* = 0.0209). In NET patients, there was a statistically significant higher prevalence of GG homozygotes of variant rs1800872 (*p* = 0.003). There was a statistically significant correlation between the rs3731198 variant and lymph node metastases (*p* = 0.0038 with Bonferroni correction). Conclusions: Our study indicates that GG homozygotes of variant rs1800872 are more often observed in NET patients, while TT homozygotes of variant rs3731198 are less frequent in this group. The rs3731198 variant may be related to an increased risk of lymph node metastasis. Further, larger multicenter studies are warranted to evaluate the potential genetic factors of sporadic NETs.

## 1. Introduction

The genetic basis of neuroendocrine tumors (NETs), despite the rapid increase in data in recent years, is still not fully understood. The majority of NETs are sporadic, although there are well-known genetic syndromes coexisting with NETs, such as multiple endocrine neoplasia type 1 (MEN1), multiple endocrine neoplasia type 4 (MEN4), von Hippel–Lindau (VHL), neurofibromatosis type 1 (NF1), and tuberous sclerosis complex (TSC), but they constitute merely a small percentage of all NETs. Although a hereditary background should be considered in many types of NETs, especially in pancreatic ones [1,2], only about 10% are associated with inherited genetic syndromes [3].

As the incidence of NETs continues to rise and the age-adjusted incidence rate has increased 6.4 times from 1973 (1.09 per 100,000) to 2012 (6.98 per 100,000) [4], it is crucial to better understand molecular alterations to more precisely assess the risk and predict the course of disease, which should significantly improve treatment effects and influence overall outcome [5].

To date, the associations of genetic alterations involved in inflammation and apoptosis have been confirmed for many malignancies, e.g., breast or colorectal cancer [6,7,8,9]. In the case of NETs, its relatively low incidence has been a certain limitation for population-wide studies on genetic alterations in NETs, but some data from literature show some of them as potentially related to NETs [10,11,12]. However, in the paper by Bodei et al., SNPs were mentioned as potential candidate factors for PRRT-related toxicity, which suggest that research on SNPs in NETs may have a practical application in the future [13]. For this reason, we analyzed selected genetic variants that may have a potential role in susceptibility for NETs and then we related them to clinical outcomes to find a potentially useful tool for the early diagnosis and optimization of NET treatment.

## 2. Materials and Methods

The study was divided into two parts: an analysis of selected (described below) genetic variants in the study and the control group and an analysis of these polymorphisms in the extended group of patients with NET in order to check whether any of those variants may affect clinical data. The clinical and demographic data of patients and controls were obtained from hospital records. All participants were of Polish origin.

The study involved the analysis of genetic material in terms of linking four germline SNPs, selected on the basis of data from the literature, with NET. The investigated material was DNA isolated from the whole peripheral blood of the participants, collected into EDTA tubes using the NucleoSpin Blood Kit (Macherey-Nagel, Düren, Germany). All patients were tested for the selected SNPs with the appropriate TaqMan probes and TaqMan Genotyping Master Mix (ThermoFisher Scientific, Waltham, USA). The reactions were prepared according to the manufacturer’s recommendations, read on a Mastercycler Realplex 2 thermocycler (Eppendorf, Hamburg, Germany) and analyzed using the incorporated data collection and analysis software (Eppendorf, Hamburg, Germany). All genetic tests were carried out in the Genetics Laboratory of the Endocrinology Clinic at the University Hospital in Krakow.

The molecular variants selected on the basis on the data from literature [11,12,13] included: rs8005354 (*DAD1*, NM_001344 intronic T/C substitution), rs2069762 (A/C substitution in the promoter region of the *IL2* NM_000586), rs3731198 (*CDKN2A*, NM_000077 intronic T/C substitution), and rs1800872 (G/T substitution in the promoter region of the *IL10* NM_000572). All patients in the control group took part in a detailed interview and physical examination to exclude a history of NET in the participant and their families, as well as other syndromes that may be part of the MEN1 syndrome: hyperparathyroidism, pituitary tumor, adrenal gland tumor; or part of VHL: kidney cancer, phaeochromocytoma, or cerebral hemangiomas. Additionally, in this group, abdominal ultrasounds were performed.

In the first part of the study, the associations of the aforementioned variants with neuroendocrine tumors were assessed and compared with those in the control group and, subsequently, with the literature data for the European population. An additional analysis was carried out for homozygotes in order to verify whether the lack of any allele (less frequent or more frequent) is favorable or unfavorable in relation to the disease. The analysis of homozygotes is synonymous with the analysis of the carriers of the other allele, regardless of their zygosity. In the next stage, molecular variants were analyzed according to the primary site, distant metastasis, and OS.

In the second part of the study, the group of patients with NET was extended. The inclusion and exclusion criteria were the same as for the first case–control part of the study. In this extended study group, the same four SNPs were assessed, and all molecular determinations were performed analogously to the procedure described above.

Next, the associations between genetic variants and clinical data (primary site, serum CgA concentration, and metastases) were analyzed.

### 2.1. Ethical Issues

This study was carried out in compliance with the 1964 Helsinki Declaration and its later amendments. All participants signed an informed consent upon enrolment (an additional informed consent was obtained for molecular analysis). The study was approved by the Ethics Committee of the Jagiellonian University Medical College: approval no. 1072.6120.250.2018.

### 2.2. Statistical Analysis

Statistical analyses were performed by use of Statistica v13 (TIBCO Software Inc., Palo Alto, CA, USA). Continuous data with normal distribution were presented as a mean value and standard deviation (SD). Categorical data were presented as percentages. Appropriate parametric or non-parametric tests were used for comparisons. OS, defined as the time from diagnosis to death due to NET or the last known date alive, was evaluated using the log-rank test. A *p* value less than 0.05 was considered statistically significant, with Bonferroni corrections applied where appropriate.

## 3. Results

### 3.1. Study Population

A total number of 185 subjects recruited from our clinical center were enrolled in this study.

### 3.2. Case–Control Part of the Study

A total of 107 patients were enrolled in the case–control part of the study. The study group comprised 49 adult Caucasian patients with NET of different primary sites. The patients were selected randomly from a cohort of about 600 NET patients (without other concomitant malignancies, positive family history of GEP NETs, and diagnosed MEN or VHL syndrome) treated or followed up in our center.

The control group consisted of 58 unrelated adult patients (patients of our center, their families and friends, as well as staff and students) with no history of NET in the themselves and their families, as well as other syndromes that may be part of the MEN1 syndrome: hyperparathyroidism, pituitary tumor, adrenal gland tumor; or in the composition of VHL: kidney cancer, phaeochromocytoma, or cerebral hemangiomas.

In the NET group, there were 46 patients diagnosed with GEP NET and 3 patients with NET of unknown primary site. The diagnosis of NET was established on the basis of the histopathological report according to the WHO classification (2010, 2017, or 2019) depending on the time of diagnosis. Histopathological findings were confirmed locally by pathologists with NET expertise.

In 49 patients, the primary site, degree of histopathological differentiation, presence of distant metastases, serum chromogranin A (CgA) concentration, and overall survival (OS) were established. The distribution of NETs according to the primary site is presented in Table 1.

The study group of NET patients involved 28 women (57%) aged 62.89 ± 9.77 years, and 21 men (43%) aged 62.10 ± 13.31 years. Distant metastases were observed in 67.34% of the patients, liver metastases were diagnosed in 57.14%, lymph node metastases in 44.9%, bone metastases in 28.5%, and peritoneum metastases in 4.1%. The median serum CgA concentration was 7.79 nmol/L (IQR = 21.6). The median overall survival (OS) at study time was 10 years (IQR = 6.0 years).

In 27 patients, the degree of histopathological differentiation was known. There were 15 (55.6%) patients with G1 tumors, 10 (37.0%) with G2, 1 case (3.7%) of G3 NET, and 1 case of NEC, which was excluded from further analyses.

The control group consisted of 58 unrelated adult patients—41 women aged 47.00 ± 13.77 years and 17 men aged 42.54 ± 16.53 years.

Of the four variants analyzed, the minor allele of the rs3731198 variant showed a trend toward a higher prevalence in the population of patients with NET, which was, however, not statistically significant, *p* = 0.0587. The minor allele of the rs1800872 variant showed a trend toward a lower incidence in NET patients compared to that in the control group, *p* = 0.0869. On the basis of the obtained results, for each variant, the minimal number of patients that would be required to obtain a statistically significant result of satisfactory power (minimum 80%) was calculated (Table 2).

The analysis of homozygotes revealed a statistically significant higher prevalence of TT homozygotes of the rs3731198 variant in the control group with *p* = 0.0209 and a power of 64.52%. The group of NET patients had a statistically significant higher prevalence of GG homozygotes of the rs1800872 variant with *p* = 0.0300 and a power of 58.43%. Both correlations were, however, non-significant after the implementation of the Bonferroni correction. The analysis of homozygotes is summarized in Table 3.

There was no statistically significant correlation of the studied variants with the primary focus or distant metastases.

### 3.3. Analysis of the Extended NET Patient Group

This part of the study involved 127 patients with the diagnosis of NET with a different primary site, established on the basis of the histopathological report according to the WHO classification (2010, 2017, or 2019) depending on the time of diagnosis. Histopathological results were also confirmed locally by pathologists with an experience in the field of NETs. The primary site, degree of histopathological differentiation, presence of distant metastases, serum CgA concentration, and OS were established. The distribution of NETs according to the primary site is presented in Table 4.

The group included 68 women (54%) and 59 men (46%). The mean age was 64.0 ± 11.4 years (in the group of women, the mean age was 65.1 ± 10.2 years; in the group of men, it was 62.6 ± 12.7 years). Distant metastases were present in 84.5% of the patients, liver metastases were confirmed in 73.4%, lymph node metastases in 55.9%, and bone metastases in 18.1%.

The median CgA serum concentration was 8.5 nmol/L (IQR = 26.6). There were no differences between the group of women and men: *p* = 0.6098. In total, 90 patients (70.9%), 53 women and 37 men, were alive at the time of the study; 36 patients (28.3%), 14 women and 22 men, were deceased, and deaths were more frequent among men (*p* = 0.0472). The median OS at study time was 7 years (IQR = 5), 6.5 years (IQR = 5.75) in men and 7 years (IQR = 5) in women.

In 92 subjects, data on the degree of histopathological differentiation were available. There were 39 (42.4%) patients with G1 tumors, 50 (54.3%) with G2, and 3 (3.3%) cases of G3 NET.

Small-intestine NETs were more common in men: *p* = 0.0246 (Fisher’s exact two-sided test), while gastric NETs were more frequent in women: *p* = 0.0097. Gastric NETs did not result in liver metastases: *p* = 0.0001 or lymph node metastases: *p* = 0.0106, while NETs of unknown primary site metastasized to the liver: *p* = 0.0402 and small-intestine NETs gave rise to nodal metastases: *p* = 0.0024 (Fisher’s exact two-sided test).

The genetic variants assessed in NET patients are summarized in Table 5.

### 3.4. Association of Clinical Data on Genetic Variants in the Extended NET Patient Group

The variants did not differ between the sexes.

Correlation of variants with CgA: after applying the correction for multiple comparisons, none of the results were within the range of statistical significance.

There was no statistical significance between the primary focus and the number of rarer alleles.

Correlation between distant metastases and the number of rarer alleles (Table 6): there was an association between variant rs3731198 and lymph node metastasis (*p* = 0.0038 with Bonferroni correction, with a power of 59.97%).

The tested variants showed no differences in the survival test (log-rank) (Appendix A Appendix A) or in the stage of disease assessed as localized/regional/advanced (Appendix A Appendix A).

## 4. Discussion

The results of the study showed that, among the analyzed variants, the rs3731198 (*CDKN2A*) variant showed a trend toward a higher prevalence in NET patients, whereas rs1800872 (*IL10* promoter) showed a trend toward a lower prevalence. None of the other variants examined, rs8005354 (*DAD1*) and rs2069762 (*IL2* promoter), were associated with NET risk. The analysis of homozygotes involved a statistically significant higher prevalence of TT homozygotes of the rs3731198 variant in the control group. There was a statistically significant correlation between the rs3731198 variant and lymph node metastases, which can be valuable in predicting the clinical course in patients with NET. In NET patients, there was a higher prevalence of GG homozygotes of the rs1800872 variant.

Despite advances in NET knowledge, the genetic factors responsible for the development of sporadic NETs are still under investigation. Moreover, further research is needed to accurately define the genomic landscape of sporadic NETs in order to estimate possible targetable molecular alterations and to identify predictive and prognostic markers. For that reason, we aimed to investigate whether common genetic variants of the genes involved in the inflammatory response, cell cycle control, and DNA repair mechanism may contribute to the risk of sporadic NETs. Analyzed variants were selected based on literature data indicating their potential association with NET development: rs3731198 [10,12], rs8005354 [10], and rs2069762 [11] or linking chosen variants to cancer susceptibility in general: rs2069762 [14] and rs180072 [15]. Our study did not confirm any associations of the studied SNPs with the incidence of NET; however, in the rs3731198 variant, there was a trend toward a higher prevalence in patients with NET and increasing the sample size to 118 patients would probably lead to such a result. However, due to the younger age of the control group and the corresponding lower risk of disease, this result should be treated with caution. The analysis of homozygotes revealed a lower incidence of TT homozygotes of that variant in NET patients. The variant rs3731198 is located in the *CDKN2A*/*CDKN2B* gene region, which is an important tumor suppressor gene involved in cell cycle regulation. Previously published papers demonstrated that this variant is associated with the risk of pancreatic NETs—first independently [11] and then in co-analysis with other variants with linkage imbalance [13]. In our study, there was association between rs3731198 and lymph node metastasis, which remains in concordance with other studies. The study of Roy et al. indicated that alterations in *CDKN2A* may contribute to the metastasis of pancreatic NETs [16].

The rs1800872 variant in the anti-inflammatory gene *IL10* decreases gene transcription. The association of this variant with neoplasms has been the subject of several studies. In a Chinese study, *IL10* rs1800872 was associated with an increased risk of esophageal cancer [17]. A meta-analysis published in 2021 confirmed the association between the rs1800872 variant and the risk of cervical cancer [18]. Several other meta-analyses have shown the association between this variant and the risk of breast cancer [19,20]. Our study found a trend toward a lower occurrence in NET patients compared to that in the control group. In the study group, there was a higher prevalence of GG homozygotes of that variant. 

The rs8005354 is an intronic variant of the *DAD1* gene. *DAD1* has been identified as a regulator of apoptosis [21]. In animal model studies, deletion of *DAD1* in the hamster TSBN7 cell line was shown to induce apoptosis [22], and the absence of *DAD1* was associated with increased apoptosis in mice [23]. This marker has been studied in various diseases to search for connections with this gene. It has not been linked to any disease in clinical databases, but it has been correlated in the literature with sporadic NETs [11]. Kulke et al. reported that *DAD1* may play a potential role in the tumorigenesis of small-intestinal NETs [24]. Our study did not confirm any associations of this variant with the incidence of NET.

IL2 is a pro- and anti-inflammatory cytokine involved in the cellular immune response [25]. Previous studies on rs2069762, which is located in the promoter region of the *IL2* gene, have not been conclusive. Data have shown that this variant is involved in increased susceptibility to nasopharyngeal [26] and oral cancer [27], and literature data have shown that this variant was associated with gastric cancer [28] and breast cancer [6]. A study from 2021 confirmed that the *IL2* gene variants rs2069762 and rs2069763 could be involved in the development of non-Hodgkin lymphoma (NHL) [29]. On the other hand, a meta-analysis from 2015 indicated that the variant *IL2* rs2069762 was not associated with cancer risk [30]. The paper by Berković et al. showed that the *IL-2* -330 T/G variant (rs2069752) may have a potential role in susceptibility to GEP NETs [12]. Our study did not show any associations of this variant with NET incidence.

The largest genome-wide association study in a group of NET patients that has been performed until now comprised more than 800 cases and more than 4500 controls. The authors identified no potential risk associations in the cohort overall, but in the small-intestine NET subgroup, they identified risk associations with three SNPs on chromosome 12 [31]. There are public databases that link genetic and clinical data, such as NCBI ClinVar, but they mainly only consider the association of genetic variants with the risk of the disease or response to treatment but not the course of the disease. For that reason, at this moment, to our best knowledge, none of them gives a possibility to verify our results.

The main limitation of our study is the small sample size. It should be noted that, due to the limited number of study participants, the power of our analysis was only about 60%. In addition, the varied number of patients, for whom good-quality genotyping results were obtained, might have also influenced the *p* value. Therefore, it would be valuable in the future to analyze those variants in additional patient groups and to include these results in meta-analyses. Due to the rarity of NETs, other larger multicenter studies are warranted to evaluate the potential genetic factors of sporadic NETs.

## 5. Conclusions

Among the tested SNPs, the rs3731198 variant showed the highest prevalence in NET patients.

Presence of the rs3731198 variant was associated with the highest risk of lymph node metastases. GG homozygosity of the rs1800872 variant was associated with a higher incidence of NET, while TT homozygosity was rarest in this group. The rarity of patients with NETs did not allow us to achieve a sufficient power value; however, our results might serve as an indication for future studies on this topic.

## Figures and Tables

**Table 1 jcm-11-05536-t001:** Distribution of primary tumor locations in the group of 49 patients analyzed in the case–control part of the study.

Site	Number (All = 49)	Percent (%)
Small intestine	14	28.6
Pancreas	12	24.5
Large intestine	12	24.5
Stomach	8	16.3
Unknown	3	6.1

**Table 2 jcm-11-05536-t002:** Frequency of alleles in the study group, the control group, and the European population according to the Genome Aggregation Database (gnomAD) v2.1.1 for the European (non-Finnish) population.

Variant	MAF Study Group [%]	MAF Control Group [%]	MAF European Population [%]	*p* Value	Minimal No. of Patients Required to Obtain a Statistically Significant Result of a Minimum 80% Power
rs8005354 (T > C)	42.86	33.67	37.60	0.1618	220
rs2069762 (A > C)	29.17	22.45	30.08	0.2615	333
rs3731198 (T > C)	15.08	7.00	13.17	0.0587	118
rs1800872 (G > T)	18.33	28.26	23.54	0.0869	142

MAF: minor allele frequency.

**Table 3 jcm-11-05536-t003:** Analysis of homozygotes.

Variant	Homozygotes of Less Frequent Allele				*p* Value	Homozygotes of More Frequent Allele				*p* Value
	Study Group		Control Group			Study Group		Control Group		
	Genotype	Frequency (%)	Genotype	Frequency (%)		Genotype	Frequency (%)	Genotype	Frequency (%)	
rs8005354 (T/C)	CC	12.7	CC	8.16	0.4414	TT	26.98	TT	40.82	0.4414
rs2069762 (A/C)	CC	5.00	CC	6.12	0.2615	AA	46.67	AA	61.22	0.1297
rs3731198 (T/C)	CC	0.00	CC	2.00	0.2595	TT	69.84	TT	88.00	0.0209
rs1800872 (G/T)	TT	3.33	TT	2.17	0.7213	GG	66.67	GG	45.65	0.0300

**Table 4 jcm-11-05536-t004:** Distribution of primary tumor locations in the group of 127 patients analyzed in the main part of the study.

Site	Number (All = 127)	Percent (%)
Small intestine	44	34.6
Pancreas	37	29.1
Unknown	20	15.8
Large intestine	15	11.8
Stomach	11	8.7

**Table 5 jcm-11-05536-t005:** Genetic variants assessed in NET patients.

Variant	No. of Genotyped Patients	Homozygote: Reference Allele	Heterozygote	Homozygote: Alternative Allele	No. of Patients with Alternative Allele
rs8005354	122 (96.1%)	T/T	T/C	C/C	81 (63.8%)
		41 (32.3%)	69 (54.3%)	12 (9.4%)	
rs2069762	122 (96.1%)	A/A	A/C	C/C	69 (54.3%)
		53 (41.7%)	60 (47.2%)	9 (7.1%)	
rs3731198	126 (99.2%)	T/T	T/C	C/C	33 (26.0%)
		93 (73.2%)	32 (25.2%)	1 (0.8%)	
rs1800872	121 (95.3%)	G/G	G/T	T/T	68 (53.5%)
		53 (41.7%)	61 (48.0%)	7 (5.5%)	

**Table 6 jcm-11-05536-t006:** Correlation between the distant metastasis and the number of rarer alleles.

Variant; More Frequent > Rarer Allele	Liver Metastases *p* Value	Lymph Node Metastases *p* Value	Bone Metastases *p* Value
rs8005354; T > C	0.7990	0.8918	0.9104
rs2069762; A > C	0.7310	0.1940	0.8124
rs3731198; T > C	0.1008	0.0038	0.8028
rs1800872; G > T	0.3022	0.7918	0.9484

## Data Availability

The data presented in this study are available on request from the corresponding author.

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
