# Peer review of "Prevalence of Selected Single-Nucleotide Variants in Patients with Neuroendocrine Tumors—Potential Clinical Relevance"

_jcm, 2022, doi:10.3390/jcm11195536_

Round 1

Reviewer 1 Report

In this manuscript, the authors studied four single nucleotide variants in NET patients. Though the project is important to improve our understanding of NET with potential clinical insights, the manuscript is suggested to be revised before publication.

First, the manuscript has writing errors and is suggested to be carefully reviewed before submission. For example, there are repeats found in abstract line 22-26, 41-44. Also, missing or extra spaces in line 34, 37, 38, 245 and 265.

Second, the study is based on the results of four variants. However, the selection of the four variants is not well explained. The rationale of the focus on the four targets should be better emphasized with discussions on the comparison of the previous studies, and potential functions of these variants in NET.

Third, the total number of the patient group is small. In this case, the p value may be influenced by the different portion of the patients that are included in the study. The authors should provide further discussion on the limitation of the study for the conclusions.

Author Response

1. In this manuscript, the authors studied four single nucleotide variants in NET patients. Though the project is important to improve our understanding of NET with potential clinical insights, the manuscript is suggested to be revised before publication.

Thank you very much for your comments concerning the article. Following your suggestions we corrected the text closely to your comments.

2. First, the manuscript has writing errors and is suggested to be carefully reviewed before submission. For example, there are repeats found in abstract line 22-26, 41-44. Also, missing or extra spaces in line 34, 37, 38, 245 and 265.

Thank you for your comment.  All noted editing errors have been corrected.

3. Second, the study is based on the results of four variants. However, the selection of the four variants is not well explained. The rationale of the focus on the four targets should be better emphasized with discussions on the comparison of the previous studies, and potential functions of these variants in NET.

Thank you for your remark.  According to your suggestion we have added the requested information highlighting the rationale for selection of used genetic variants including additional references. Furthermore we have compared in more details our results with data available in literature.

4. Third, the total number of the patient group is small. In this case, the p value may be influenced by the different portion of the patients that are included in the study. The authors should provide further discussion on the limitation of the study for the conclusions.

As suggested, we have modified the discussion and conclusions to highlight the small number of patients as one of the most important limitations of the study.

Reviewer 2 Report

The article entitled "Prevalence of selected single nucleotide variants in patients with neuroendocrine tumours - potential clinical relevance" presents the analysis of rs8005354 (DAD1, NM_001344 intronic T/C substitution), rs2069762 (T/G substitution in the promoter region of IL2 NM_000586), rs3731198 (CDKN2A, 24 NM_000077 intronic A/G substitution) and rs1800872 (C/A substitution in the promoter region of 25 IL10 NM_000572) in healthy controls and NET patients. The study has many weaknesses: the primary one is a difference of approximately two decades between the mean age of controls and cases. The authors are unable to make a case for the statistical soundness of the analysis - insufficient power of the study. The manuscript needs to be thoroughly revised for language to ensure preciseness and scientific clarity. In the present form, the work is not of sufficient merit to recommend publication.   

Author Response

1. The article entitled "Prevalence of selected single nucleotide variants in patients with neuroendocrine tumours - potential clinical relevance" presents the analysis of rs8005354 (DAD1, NM_001344 intronic T/C substitution), rs2069762 (T/G substitution in the promoter region of IL2 NM_000586), rs3731198 (CDKN2A, 24 NM_000077 intronic A/G substitution) and rs1800872 (C/A substitution in the promoter region of 25 IL10 NM_000572) in healthy controls and NET patients. The study has many weaknesses: the primary one is a difference of approximately two decades between the mean age of controls and cases. The authors are unable to make a case for the statistical soundness of the analysis - insufficient power of the study.

We agree that a difference of approximately two decades between the mean age of controls and cases is a limitation of the study. Therefore, we also considered in the analysis (the general population from the available database) the frequency of alleles in the European population, according to the Genome Aggregation Database (gnomAD) v2.1.1 for the European (non-Finnish) population. In addition, being aware of this limitation, we draw conclusions mainly on the differences in disease course within the NET group. To be more precise we have added a relevant comment.

2. The manuscript needs to be thoroughly revised for language to ensure preciseness and scientific clarity. In the present form, the work is not of sufficient merit to recommend publication.

The manuscript has been revised by a native speaker.

We hope you will find our clarifications sufficient and the corrected manuscript suitable for publication.

Reviewer 3 Report

The authors gave a well description of prevalence of selected single nucleotide variants in patients with neuroendocrine tumours (NETs) - potential clinical relevance in the present study, which has significance to evaluate the potential genetic factors of NETs. However, my major question is that the sample is small and statistical power may be affected in the paper. In the discussion, the authors should mention the limitation of methods.

Author Response

The authors gave a well description of prevalence of selected single nucleotide variants in patients with neuroendocrine tumours (NETs) - potential clinical relevance in the present study, which has significance to evaluate the potential genetic factors of NETs. However, my major question is that the sample is small and statistical power may be affected in the paper. In the discussion, the authors should mention the limitation of methods. 

Thank you for your comments. As suggested, we have extended the discussion section by relevant information about the limitation of used methods including small sample size and obtained statistical power of the study.

Reviewer 4 Report

The authors presented a very interesting report on prevalence of selected single nucleotide variants in patients with NETs. The rarity of NETs makes these results valuable for understanding their pathogenesis. The manuscript is well prepared and does not rise any major concerns. I have only a few minor remarks:

1.       I believe that p representing statistical significance should be written in italics (e.g. p = 0.05).

2.       Please modify table captions as both Table 1 and 2 have the same one.

3.       The manuscript should be checked by a native speaker.

Author Response

The authors presented a very interesting report on prevalence of selected single nucleotide variants in patients with NETs. The rarity of NETs makes these results valuable for understanding their pathogenesis. The manuscript is well prepared and does not rise any major concerns. I have only a few minor remarks:

  1. I believe that prepresenting statistical significance should be written in italics (e.g. p = 0.05).
  2. Please modify table captions as both Table 1 and 2 have the same one.
  3. The manuscript should be checked by a native speaker.

Thank you very much for your comments concerning the article.

Following your suggestions we corrected the text including modification of Table 1 and 2 captions and form of writing p representing statistical significance . The manuscript has also been revised by a native speaker.